# Traffic-originated nanocluster emission exceeds $H_2SO_4$-driven photochemical new particle formation in an urban area

Miska Olin[1], Heino Kuuluvainen[1], Minna Aurela[2], Joni Kalliokoski[1], Niina Kuittinen[1], Mia Isotalo[1], Hilkka J. Timonen[2], Jarkko V. Niemi[3], Topi Rönkkö[1], and Miikka Dal Maso[1]

[1]Aerosol Physics Laboratory, Physics Unit, Tampere University, FI-33014 Tampere, Finland
[2]Atmospheric Composition Research, Finnish Meteorological Institute, FI-00101 Helsinki, Finland
[3]Helsinki Region Environmental Services Authority (HSY), FI-00066 HSY, Finland
**Correspondence:** Miska Olin (miska.olin@tuni.fi)

**Abstract.** Elevated ambient concentrations of sub-3 nm particles (nanocluster aerosol, NCA) are generally related to atmospheric new particle formation events, usually linked with gaseous sulfuric acid ($H_2SO_4$) produced via photochemical oxidation of sulfur dioxide. According to our measurement results of $H_2SO_4$ and NCA concentrations, traffic density, and solar irradiance at an urban traffic site in Helsinki, Finland, the view of aerosol formation in traffic-influenced environments is updated by presenting two separate and independent pathways of traffic affecting the atmospheric NCA concentrations: by acting as a direct nanocluster source, and by influencing the production of $H_2SO_4$. As traffic density in many areas is generally correlated with solar radiation, it is likely that the influence of traffic-related nanoclusters has been hidden in the diurnal variation, and is thus underestimated because new particle formation events also follow the diurnal cycle of sunlight. Urban aerosol formation studies should, therefore, be updated to include the proposed formation mechanisms. The formation of $H_2SO_4$ in urban environments is here separated in two routes: primary $H_2SO_4$ is formed in hot vehicle exhaust and is converted rapidly to particle phase; secondary $H_2SO_4$ results from the combined effect of emitted gaseous precursors and available solar radiation. A rough estimation demonstrates that $\sim 85\%$ of the total NCA and $\sim 68\%$ of the total $H_2SO_4$ in urban air at noontime at the measurement site are contributed by traffic, indicating the importance of traffic emissions.

## 1 Introduction

Urban environments exhibit some of the highest aerosol particle concentrations encountered in the Earth's atmosphere. Elevated particle concentrations are related to adverse health effects (Dockery et al., 1993; Pope et al., 2002; Beelen et al., 2014) and various effects on climate (Arneth et al., 2009). Recent studies on urban aerosol particles have focused attention on the formation process of sub-3 nm particles (Zhao et al., 2011; Kulmala et al., 2013; Kontkanen et al., 2017) also called nanocluster aerosol (NCA) (Rönkkö et al., 2017). The importance of photochemical formation mechanisms, involving, e.g., sulfuric acid ($H_2SO_4$) and ammonia (Yao et al., 2018) or other photochemically produced vapors (Lehtipalo et al., 2018), has

been highlighted. However, these studies omit the important role of direct emission of NCA-sized particles in their analysis, despite recent findings that traffic is a major source of such particles (Rönkkö et al., 2017). The proposed mechanisms also assume that key precursor vapors are formed via photochemical oxidation (Paasonen et al., 2010; Lehtipalo et al., 2018).

The most important gaseous species forming new particles in the atmosphere is $H_2SO_4$, the main source of which is usually considered to be sulfur dioxide ($SO_2$). $SO_2$ is photochemically oxidized in the atmosphere by an oxidizing agent produced by solar radiation, such as hydroxyl radical (OH) (Kulmala et al., 2014). $H_2SO_4$ produced via this route is here termed *secondary $H_2SO_4$*. Sources of regional $SO_2$ include shipping, power generation, atmospheric oxidation of dimethyl sulfide, and volcanic activity. Additionally, motor vehicles emit $SO_2$ due to sulfur-containing fuels and lubricant oils; hence, also traffic can contribute to the secondary $H_2SO_4$ levels. A part of $SO_2$ formed during combustion is oxidized to $H_2SO_4$ already in vehicles' oxidative exhaust after-treatment systems (Arnold et al., 2012), which makes vehicles direct $H_2SO_4$ emitters. In contrast to the secondary $H_2SO_4$ formed via photochemistry, $H_2SO_4$ formed in hot exhaust without the need of solar radiation is here termed *primary $H_2SO_4$*. In principle, primary $H_2SO_4$ can also contribute to the atmospheric $H_2SO_4$ concentrations, at least in the vicinity of traffic.

Ambient aerosol particles are either emitted directly into the atmosphere as primary particles or new particles are formed from atmospheric precursor gases in a new particle formation (NPF) process. NPF processes have been shown to occur in a variety of environments, and their occurrence is believed to be controlled, on one hand, by the availability of particle-forming vapors and, on the other hand, by the reduction of the vapors and fresh cluster-sized nuclei due to pre-existing aerosol surface area acting as a condensation sink (CS) (McMurry and Friedlander, 1979; Kerminen et al., 2018). The observations that many urban areas display high numbers of NCA particles have been puzzling because aerosol in this size range has generally been associated with NPF processes, which are unexpected due to high CS in urban areas.

Simultaneously, evidence has been mounting that the exhaust of road vehicles often contains high numbers of particles in the nucleation mode size range (5–50 nm) (Kittelson, 1998) and, recently, that traffic is a direct source of NCA-sized particles (Rönkkö et al., 2017). A recent study by Yao et al. (2018) presented data of high NCA concentrations in a highly polluted urban area, with an interpretation that they are formed in a regional NPF process. Here, this view is contrasted and complemented by presenting data from a one-month measurement campaign performed in May 2017 at curbside of a densely trafficked street in an urban area of Helsinki, Finland. NCA concentration data from this curbside measurement have already been analyzed by Hietikko et al. (2018) with the conclusion of traffic inducing a dominant signal on NCA concentrations, according to diurnal variation and wind direction. Here, we extend the analysis with the data of $H_2SO_4$ concentrations and solar irradiances (SI) to distinguish interfering processes of traffic and regional NPF on NCA concentrations. The $H_2SO_4$ measurement at the curbside connects urban $H_2SO_4$ concentrations to traffic sources quantitatively, for the first time. The data provide reference data for primary $H_2SO_4$ emission data and the ability to determine emission factors of vehicles in a real-world driving situation.

Prescribed primary emissions in current regional and global aerosol models do not include particles in the smallest size range (Paasonen et al., 2016). The NCA-sized particle concentrations in models are therefore mainly driven by photochemical NPF processes, omitting the directly emitted NCA-sized particles. Due to the unknown chemical composition of the traffic-originated NCA-sized particles, significant NCA-related health risks cannot be excluded. Especially for solid NCA,

their behavior inside the body, such as penetrating directly into brains through the olfactory nerve (Maher et al., 2016), may have hitherto unknown adverse effects. Measuring the composition of NCA particles directly is very challenging with current technologies due to very small particle size and thus very low mass of NCA particles. An alternative way to study particle composition is to study the formation mechanism of the particles, which is one objective of this study.

## 2 Methods

### 2.1 Measurement site

The measurement site was located at a street canyon at Mäkelänkatu about 3 km north of the city center of Helsinki, Finland (Fig. 1). The devices for gas and aerosol measurements were installed in two containers next to each other (Fig. 2) located at the curbside of the street canyon.

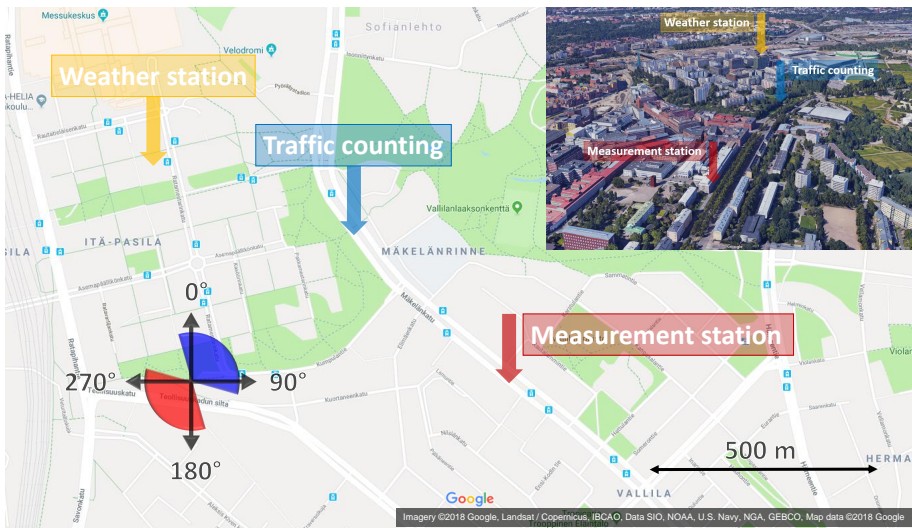

**Figure 1.** The map of the measurement site. Red and blue sectors denote the wind directions which result in the flow field coming from the street direction and from the background direction, respectively, towards the measurement container.

Traffic count was measured in 15 min time resolution by the City of Helsinki at the same street but 600 m north of the measurement containers. Environmental parameters, such as wind velocity, wind direction, SI, air temperature, pressure, relative humidity, and precipitation, were measured at a weather station on the rooftop of a 53 m high building 900 m northwest of the measurement containers. The location of the weather station provided measurement data which are undisturbed by other buildings but the location was still sufficiently near to the measurement containers to provide representative values.

The street canyon consists of three lanes for cars to both directions, two rows of trees, two tramlines, and two pavements, resulting in total width of 42 m and height of 17 m (Kuuluvainen et al., 2018). Due to the vortex affecting the flow field in a street canyon, the wind direction at the measurement containers was considered opposite to the direction above the roofs

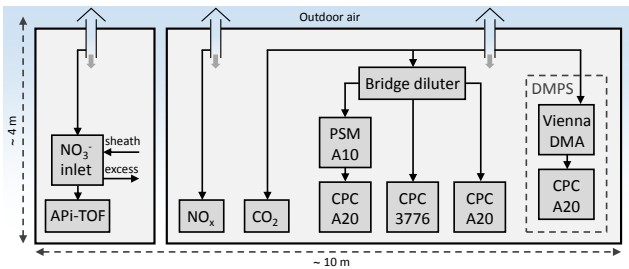

**Figure 2.** The measurement setup inside the containers at the curbside of the street canyon.

(Ahmad et al., 2005). Therefore, the wind direction diagram in Fig. 1 is mirrored by the street canyon axis. However, the street canyon in this case is not a regular street canyon but a wide avenue canyon and it has a displaced building near the measurement location. This can cause some errors to the actual flow field interpreted using the mirrored wind direction, and the effect of wind direction on the measured emissions is not seen as clearly as in an open environment or in a regular street canyon.

The measurement setup inside the containers is shown in Fig. 2. Outdoor air samples to the measurement devices were drawn through the roof of the containers 4 m above the ground, using vertical probes having diameters of 50 mm and flow rates higher than 200 lpm to minimize losses onto the walls of the sampling lines.

## 2.2   Sulfuric acid measurements

$H_2SO_4$ was measured in the gas phase using a nitrate-ion-based ($NO_3^-$-based) chemical ionization atmospheric pressure in-
10 terface time-of-flight mass spectrometer (CI-APi-TOF; Aerodyne Research Inc.; USA and Tofwerk AG, Switzerland; Jokinen et al., 2012). It consists of a chemical ionization (CI) inlet (Eisele and Tanner, 1993) and an APi-TOF mass spectrometer (Junninen et al., 2010).

     The CI-inlet was operated by ionizing small concentration of nitric acid ($HNO_3$) vapor in the sheath air using X-ray to produce $NO_3^-$ ions. The sheath air flow rate to the CI-inlet was 20 lpm and it was generated in two ways: during the first
two weeks, a small pump followed by an HEPA filter was used; and during the last two weeks, an oil-lubricated compressor followed by a coarse particle, an oil, a water droplet, an HEPA, and an activated carbon filters was used. The excess flow from the CI-inlet using a vacuum pump had a flow rate of 30 lpm, resulting in the sample flow rate of 10 lpm to the inlet.

     $H_2SO_4$ is detected in the CI-APi-TOF as bisulfate ions ($HSO_4^-$) and as $HSO_4^-$ ions clustered with $HNO_3$ through the following reactions:

$$H_2SO_4 + NO_3^- \cdot (HNO_3)_n \rightarrow HSO_4^- + (n+1)HNO_3 \tag{R1}$$

$$H_2SO_4 + NO_3^- \cdot (HNO_3)_n \rightarrow HSO_4^- \cdot HNO_3 + nHNO_3 \tag{R2}$$

where $n = 0, 1, ...$ The $H_2SO_4$ concentrations are calculated with the equation:

$$[H_2SO_4] = \tag{1}$$

$$\frac{C}{P} \cdot \frac{\{HSO_4^-\} + \{HSO_4^- \cdot HNO_3\} + \{HSO_4^- \cdot H_2SO_4\}}{\{NO_3^-\} + \{NO_3^- \cdot HNO_3\} + \{NO_3^- \cdot (HNO_3)_2\}}$$

where $C$ is the calibration coefficient for $H_2SO_4$, $P$ is the penetration efficiency of $H_2SO_4$ in the sampling lines, and the {}-braces denote the areas of the peaks at corresponding mass-to-charge ratios in the high-resolution spectra measured by the TOF mass spectrometer. The calibration coefficient was determined by generating known concentrations of $H_2SO_4$ using the oxidation of $SO_2$ by OH radical (Kürten et al., 2012). The calibration coefficient was determined for the both sheath air generations: the values are $C = 1.3 \times 10^9\,\text{cm}^{-3}$ for the pump-based sheath air and $C = 9.1 \times 10^9\,\text{cm}^{-3}$ for the compressor-based sheath air. The values differ due to different purities of the sheath air.

The diffusional losses (Brockmann, 2005) of $H_2SO_4$ in the sampling lines were calculated with the diffusion coefficient of $0.071\,\text{cm}^2/\text{s}$ representing the diffusion coefficient of a hydrated $H_2SO_4$ molecule in the relative humidity of 60 % and temperature of 283 K (Chapman and Cowling, 1954; Hanson and Eisele, 2000). The calculated penetrations are $P = 0.30$ when pump-based sheath air was used and $P = 0.22$ when compressor-based sheath air was used. These values differ because minor changes to the sampling lines were also done when the compressor was installed.

The $H_2SO_4$ concentrations from zero measurements are subtracted from the measured $H_2SO_4$ concentrations. The zero measurements were done by using the sheath air as a sample to obtain the $H_2SO_4$ concentration originated from the sheath air generation. The $H_2SO_4$ concentrations during the zero measurements were $3.7 \times 10^5\,\text{cm}^{-3}$ with the pump-based sheath air and $1.8 \times 10^6\,\text{cm}^{-3}$ with the compressor-based sheath air. Due to the limitations of space inside the containers, higher level of purification for the sheath air was not available.

The data from the APi-TOF was recorded with the time resolution of 2 s, but at least 1 min of the raw data is required for averaging to obtain feasible high-resolution spectra.

## 2.3 Gas measurements

Nitric oxide (NO) and nitrogen dioxide ($NO_2$) concentrations were measured using Horiba APNA-370 and the data were recorded with a time resolution of 1 min. In this study, only the sum of NO and $NO_2$ concentrations, denoted as $NO_x$ concentration, is used in the analysis. Carbon dioxide ($CO_2$) concentration was measured using LI-COR LI-7000 analyzer with a time resolution of 1 s.

Because traffic density and the concentrations of emissions at the curbside are not directly correlated due to turbulent flow field and variable wind directions causing the emissions to be diluted in a different extent at the measurement location, a traffic-originated tracer is needed to connect the observed concentrations quantitatively to traffic emissions. An ideal tracer is one that is universally emitted by all vehicles and is not altered in the atmosphere during the time scale of the exhaust plume dilution process. $CO_2$ is emitted by all combustion engines, with the emission rate proportional to the fuel consumption; thus, it is used as a tracer in determining emission factors of traffic. The drawback of $CO_2$ as a traffic tracer is its varying background concentration due to regional-level phenomena. The background concentration is also higher than the concentration increase

of traffic. As the main source of $NO_x$ in urban areas is traffic (Clapp and Jenkin, 2001), the background concentration is low and causes thus no significant uncertainty to the traffic contribution. However, the drawback of $NO_x$ is its varying emission rates across the whole vehicle fleet (Yli-Tuomi et al., 2005). Concluding, we decided to use $NO_x$ as the traffic tracer, except in the emission factor analysis where $CO_2$ is used due to its direct connection to the fuel consumption.

The $NO_x$ concentration ($[NO_x]$) correlates well with the traffic density on weekdays (Fig. 3). However, on weekends, much stronger dilution conditions during daytime compared to nighttime are seen. During nighttime, there is a peak in the $NO_x$ concentration though it is nonexistent in traffic density, which suggests stagnant weather conditions coincided at nighttime on weekends for the considered time range. The $NO_x$ concentrations were higher during the morning rush hours than during the afternoon rush hours on weekdays although the traffic density behaved oppositely, which occurs because, during the morning rush hours, traffic was concentrated on the same side of the street as the measurement containers located providing a shorter distance, and thus less dilution, for the emissions to travel to the measurement devices.

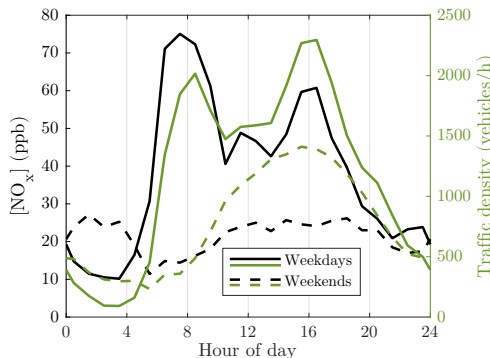

**Figure 3.** Average diurnal variations of nitrogen oxides ($NO_x$) concentration, representing the concentration of the traffic-originated emissions overall, and traffic density. The $NO_x$ concentration data measured with 1 min time resolution are firstly averaged to 1 h resolution; secondly, the averaged data from different days are averaged geometrically for the specific hours. Geometric averages are used for emissions, instead of arithmetic averages, because the logarithmic nature of the concentrations would cause skewed frequency distributions; thus, the highest concentrations would be weighted more than the lower ones. Arithmetic averaging is used for the traffic density.

## 2.4 Particle measurements

The number concentration of particles with the diameters larger than approximately 1.2 nm were measured using an Airmodus A10 Particle Size Magnifier (PSM A10) (Vanhanen et al., 2011) with a diethylene glycol saturator flow rate of 1.3 lpm followed by an Airmodus A20 Condensation Particle Counter (CPC A20). Particles larger than 3 and 7 nm were measured using a TSI 3776 Condensation Particle Counter (CPC 3776) and another Airmodus CPC A20, respectively. The particle size distribution between 6 and 800 nm was measured using a Differential Mobility Particle Sizer (DMPS) consisting of a Vienna-type Differential Mobility Analyzer (DMA) followed by a CPC A20. Due to high particle concentrations at the street canyon, the sample to CPCs was diluted using a bridge diluter having a dilution ratio of 8.2. The dilution ratio for the specific diluter is, however,

measured for larger particle sizes only, and because the diluter is based on diffusional losses of the particles, the dilution ratio for NCA-sized particles is higher. Therefore, the NCA concentrations reported here represent the lower limits of the actual concentrations.

The number concentration of nanocluster aerosol, particles within the diameter range between 1.2 and 3 nm, can be calculated by subtracting the concentration measured by the CPC 3776 from the concentration measured by the PSM. The particle size distribution between 1.2 nm and 800 nm can be calculated with the data from all these aerosol measurement devices by taking the cut diameters of the CPCs and the dilution ratio of the bridge diluter into account. The NCA concentration was measured with a time resolution of 1 s and the size distribution with a time resolution of 9 min.

## 3   Results and discussion

The data from the off-site measurements of traffic count and environmental parameters are available for the whole four-week measurement campaign starting on 4 May 2017 and ending on 31 May 2017 (see the Supplement for the time series). This time range provided adequate data for examining NCA and $H_2SO_4$ formation respect to solar irradiance, because there were sufficient amounts of days both with clear sky and with cloud cover, yet without too many rainy days. There are some gaps in the NCA and $H_2SO_4$ data during the four weeks due to unavailability of the measurement devices. The data analysis considers only the time ranges for which all the measurement data are available, resulting in three weeks of data.

Figure 4 presents our proposal for the updated mechanism of $H_2SO_4$ and particle formation in traffic-influenced areas, based on our measurement results. The most noteworthy details are illustrated with red crosses indicating $H_2SO_4$ routes which were observed to occur barely only, or not at all. As shown later in this section, our measurement at the curbside displays no clear increase in gaseous $H_2SO_4$ concentrations with increasing traffic volumes. With the fact that vehicles do emit primary $H_2SO_4$ (Arnold et al., 2012; Rönkkö et al., 2013), it is evident that the majority of primary $H_2SO_4$ must be converted to the particle phase via nucleation (route 1A) and condensation (1B) rapidly after emission. Conversely, secondary $H_2SO_4$ potentially remains longer in urban atmosphere because it does not experience conditions favoring such a rapid gas-to-particle conversion, i.e., rapid temperature decrease, high precursor concentrations, and high pre-existing CS. Therefore, the signal of $H_2SO_4$ measured from the curbside of the street is mainly due to secondary $H_2SO_4$ only.

Our results show that both traffic and regional NPF influence NCA concentrations at the urban traffic site, with direct NCA emission from traffic dominating. Comparison of the NCA and $H_2SO_4$ concentrations with SI and traffic density suggests that while solar radiation favors higher NCA concentrations, the photochemically produced $H_2SO_4$ may not be the key compound in the presence of NCA in urban areas. Traffic-originated NCA particles may be formed via a delayed primary emission route by rapid nucleation of low-volatile gaseous compounds emitted by vehicles during exhaust cooling after releasing from the tailpipe (1A). On the other hand, they may be solid particles emitted directly by engines, via a primary emission route (Sgro et al., 2012; Alanen et al., 2015). Although it is likely that both nucleation (1A) and condensation (1B) routes from primary $H_2SO_4$ exist because nucleation mode particle number concentrations and particle sizes are correlated with the $H_2SO_4$ concentration in exhaust (Arnold et al., 2012; Rönkkö et al., 2013), the ratio of the routes at our measurement site is not determined. Neither

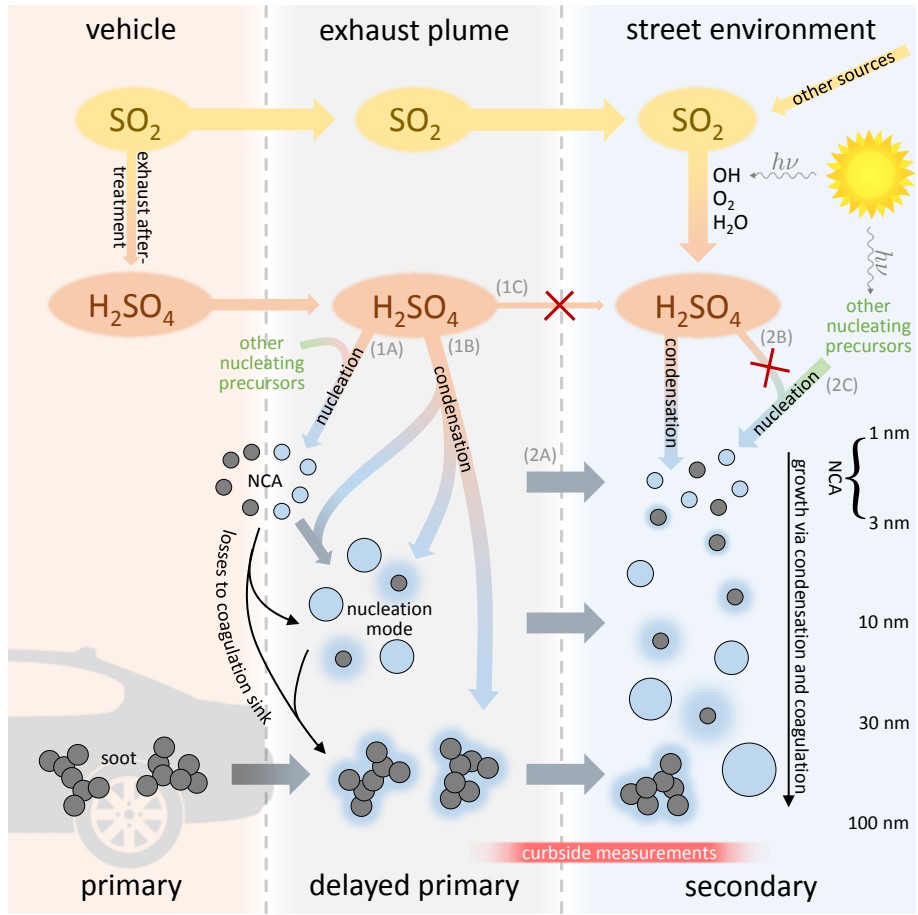

**Figure 4.** Proposed mechanism of sulfuric acid and particle formation in traffic-influenced areas. The route of primary $H_2SO_4$ to urban atmosphere (route 1C) is largely terminated to particle phase rapidly after the emission (routes 1A and 1B). Conversely, secondary $H_2SO_4$ remains in urban atmosphere because it does not experience such a rapid gas-to-particle conversion. NCA presence in urban atmosphere is majorly controlled by traffic emissions (route 2A) and only marginally by nucleation from secondary precursors (routes 2B and 2C), especially the contribution of nucleation from secondary $H_2SO_4$ (route 2B) is noticeably overridden by traffic emissions (route 2A).

is the ratio of NCA particles emitted primarily and through the nucleation route (1A) determined. Therefore, the relative proportion of $H_2SO_4$ in traffic-originated NCA particles remains unknown, leading to the possibility of solid NCA emissions.

The first evidence for traffic-contributed concentrations of NCA ($N_{NCA}$) and $H_2SO_4$ was found in the diurnal variations of the NCA, $H_2SO_4$, and $NO_x$ concentrations and SI (Fig. 5). The diurnal variations on weekdays (Fig. 5a) differ from the diurnal variations on weekends (Fig. 5b). The main difference between weekdays and weekends are traffic volumes; therefore, such a difference in the concentrations of NCA and $H_2SO_4$ should only be expected if their formation are in some manner connected to traffic. The connection of NCA to traffic is further strengthened by comparing it to the $NO_x$ concentrations, which are directly linked to traffic densities and traffic-related emissions (Fig. 3). On weekdays, the NCA concentration increased in tandem with the $NO_x$ concentration during the morning rush hours. On weekends, the NCA concentration increased at noontime much more clearly than the $NO_x$ concentration. This can be interpreted as a sign of an ongoing regional NPF process producing NCA particles with high SI. On weekdays, the regional NPF process should only produce higher NCA concentration during afternoon rush hours having higher SI compared to morning. The increased NCA concentrations during the morning rush hours suggest that the traffic-originated NCA does not require solar radiation to form.

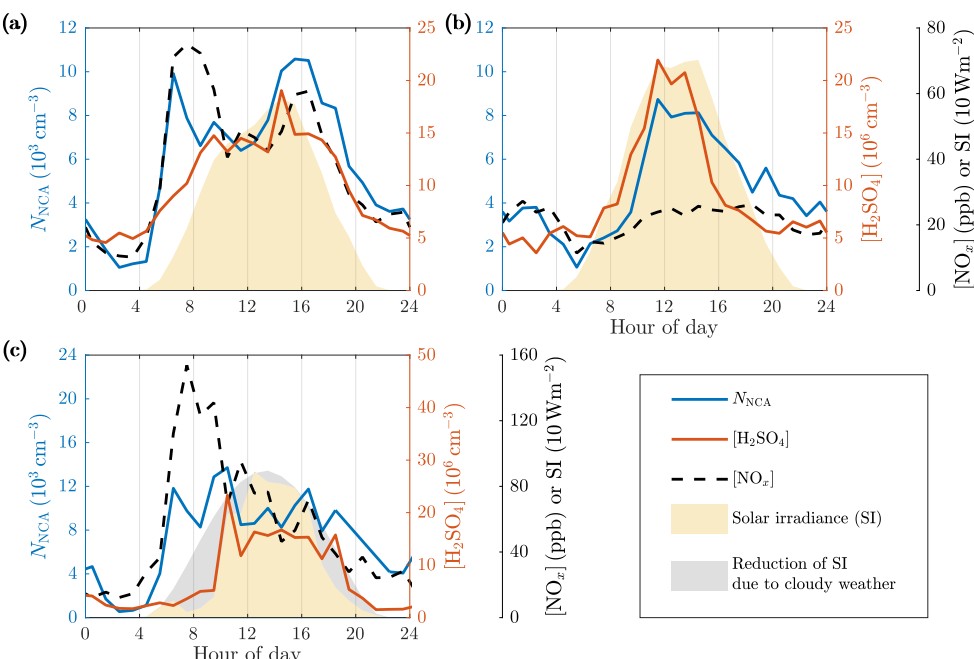

**Figure 5.** Average diurnal variations of the NCA, $H_2SO_4$, and $NO_x$ concentrations and solar irradiance (SI) on (**a**) weekdays, (**b**) weekends, and (**c**) example day 5/22/2017 with cloudy morning and evening. The concentration data firstly averaged to 1 h time resolution are averaged geometrically as described in Fig. 3.

We observed that, traffic levels influence the $H_2SO_4$ concentrations, but they are still mainly controlled by solar radiation. In contrast to the NCA concentration, the $H_2SO_4$ concentration traced SI much more closely, with a maximum at noontime and minimum at night. On weekdays, a peak in the $H_2SO_4$ concentration during afternoon rush hours suggests that traffic might also influence the formation of $H_2SO_4$. Further evidence for this is found by comparing the diurnal variation of $H_2SO_4$ between weekdays and weekends. On weekends, the $H_2SO_4$ concentration increased not until the traffic density and the $NO_x$ concentration were also increased, whereas on weekdays, the traffic density was already high when SI and the $H_2SO_4$ concentration began to increase. Furthermore, higher irradiances were required on weekends before the rise in the $H_2SO_4$ concentration and, additionally, the order of the increase in the NCA and $H_2SO_4$ concentrations was switched.

The time series show that NCA is not similarly controlled by solar radiation but rather by traffic. This is clearly showcased in Fig. 5c which presents data from a day with cloudy weather reducing SI in the morning and in the evening but still with a constant wind direction. The NCA concentration closely traced traffic levels in the morning, whereas the $H_2SO_4$ concentration only increased when SI increased hours later. This clearly shows that the formation of NCA, in this case, is independent of SI and the $H_2SO_4$ concentration. It is also noteworthy that no increase in the NCA concentration is observed when SI increased, suggesting that traffic dominated in the NCA formation. There were also other days with cloudiness decreasing SI with similar observations; however, the example day in Fig. 5c was the day with the most clear effect of cloudiness on SI and the reduction of SI coincides with morning rush hours, displaying high NCA concentrations.

The data suggest that the formation of atmospheric $H_2SO_4$ is strongly enhanced in the presence of both traffic and sunlight. While a strong correlation between the NCA and $NO_x$ concentrations (Fig. 6a, b: Pearson's $R = 0.84$) confirms the connection between NCA and traffic, a remarkably weaker, but also positive, correlation between the $H_2SO_4$ and $NO_x$ concentrations ($R = 0.50$) was observed, revealing the connection between $H_2SO_4$ and traffic. The effect of SI at different traffic densities shows differing patterns for NCA (Fig. 6c) and $H_2SO_4$ (Fig. 6d). While high SI is associated with higher NCA and $H_2SO_4$ levels, traffic density determines the base level for both (the concentrations at zero SI). For $H_2SO_4$, the influence of traffic causes a marked increase in the slope of the $H_2SO_4$ concentration-SI-line. The slope can be interpreted as the production efficiency of $H_2SO_4$ via photochemistry. It is evident that for NCA, the influence of traffic dominates in comparison to SI, as the traffic-influenced NCA concentration (red data) exceeds the non-traffic concentrations (black data) even during dark times. For $H_2SO_4$, the situation is different, as all dark-period $H_2SO_4$ concentrations are close to equal levels. These differing patterns suggest that the majority of NCA in traffic-influenced areas is formed independently of secondary $H_2SO_4$, in contrast to the findings of Yao et al. (2018).

Even more compelling evidence for traffic-originated NCA and $H_2SO_4$ can be found by comparing the observed concentrations to $CO_2$ concentrations ($[CO_2]$). In Fig. 7, no apparent difference in the emission factors of NCA for different SI are seen; however, in the case of $H_2SO_4$, higher SI lead to noticeably higher emission factors of $H_2SO_4$. We tested for potential co-correlations between SI and traffic density to examine potential traffic level increase with simultaneous SI increase due to their almost similar diurnal cycles. We found no clear correlation between $CO_2$ concentration and SI (Fig. S5). In a case of a found correlation, the slopes in Fig. 7 could not have been interpreted as emission factors, but as photochemical production due to accelerated photochemistry with higher SI values. Although the varying background concentration of $CO_2$ causes uncer-

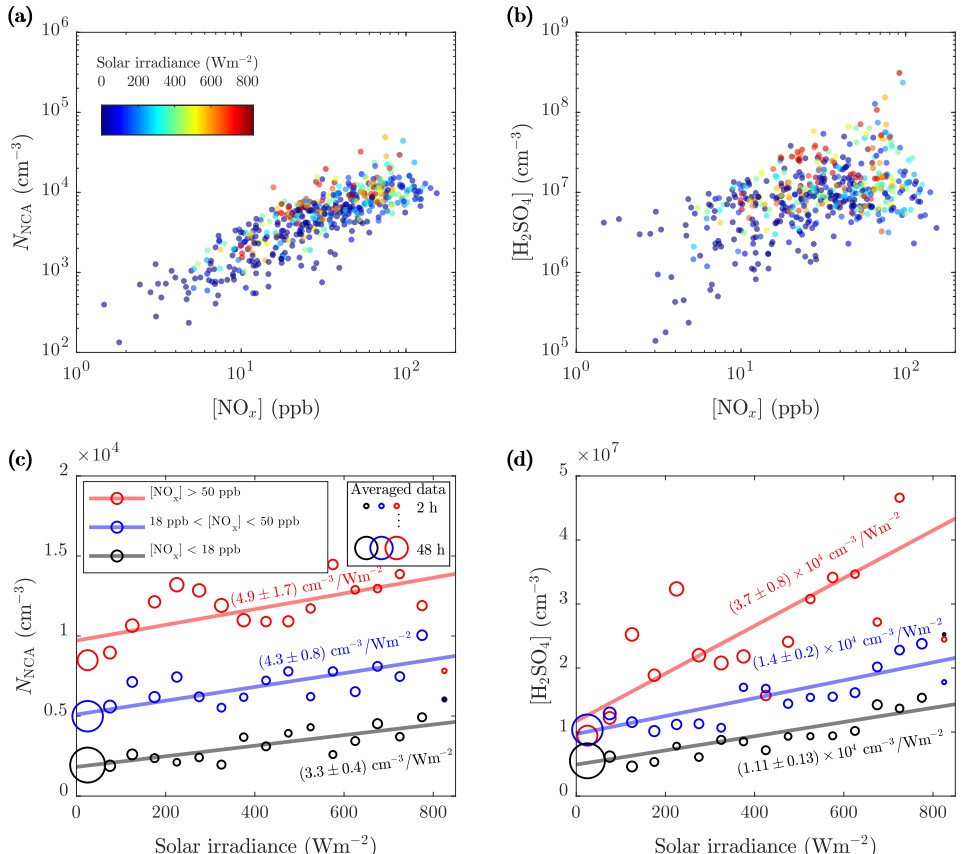

**Figure 6.** 1-hour-averages of the (**a**) NCA and (**b**) $H_2SO_4$ concentrations as a function of the $NO_x$ concentration colored by the solar irradiance (SI) and 10-min-averages of the (**c**) NCA and (**d**) $H_2SO_4$ concentrations further averaged to 17 different SI bins (bin width is chosen to provide clear graphical representation) for three $NO_x$ concentration ranges. Weighted least squares fitting, with data point count in bin-averaging (shown with the circle diameters) as weighs, for the bin-averaged values was done to output linear fits. The slopes are marked in the figure and are not largely affected by the chosen bin width. The intercepts are (**c**) $(0.182 \pm 0.013) \times 10^4 \, \text{cm}^{-3}$, $(0.51 \pm 0.03) \times 10^4 \, \text{cm}^{-3}$, and $(0.97 \pm 0.07) \times 10^4 \, \text{cm}^{-3}$; and (**d**) $(0.49 \pm 0.04) \times 10^7 \, \text{cm}^{-3}$, $(0.97 \pm 0.06) \times 10^7 \, \text{cm}^{-3}$, and $(1.2 \pm 0.3) \times 10^7 \, \text{cm}^{-3}$, for the lowest, the mid-ranged, and the highest $NO_x$ concentration range, respectively.

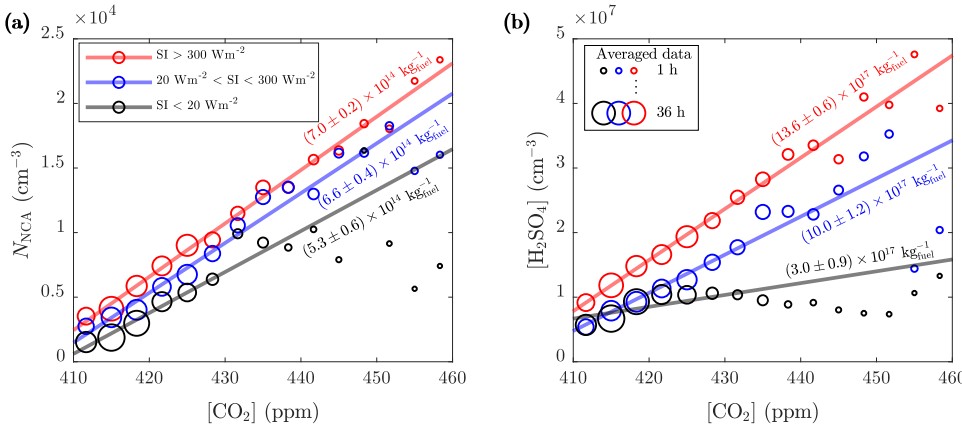

**Figure 7.** 1-min-averages of the (**a**) NCA and (**b**) $H_2SO_4$ concentrations further averaged to $CO_2$ concentration bins for three SI ranges (see Fig. 6 for the details in averaging and linear regression). The slopes of the linear fits converted to kilograms of fuel combusted (using the emission factor of $CO_2$, 3.14 kg per 1 kg of fuel combusted (Yli-Tuomi et al., 2005)) are marked in the figure. The intercepts of the fits at 410 ppm are (**a**) $(0.07 \pm 0.04) \times 10^4\,\mathrm{cm}^{-3}$, $(0.15 \pm 0.05) \times 10^4\,\mathrm{cm}^{-3}$, and $(0.25 \pm 0.03) \times 10^4\,\mathrm{cm}^{-3}$; and (**b**) $(0.67 \pm 0.07) \times 10^7\,\mathrm{cm}^{-3}$, $(0.49 \pm 0.15) \times 10^7\,\mathrm{cm}^{-3}$, and $(0.79 \pm 0.06) \times 10^7\,\mathrm{cm}^{-3}$, for the lowest, the mid-ranged, and the highest SI range, respectively.

tainty in analyzing the contribution of traffic on emissions, linear dependencies are still observed in Fig. 7. These results again support the finding that solar radiation is required for the formation of $H_2SO_4$ from traffic emissions and demonstrates clearly that both NCA and $H_2SO_4$ originate from traffic. This is further supported by examining the concentrations in different wind directions (Figs. 8 and 9) which shows that the highest concentrations were measured when the wind blew from the street.

While the emission factors can depend markedly on vehicle, engine, fuel, and after-treatment system types, the emission factors obtained here represent the average fleet-level values and can thus be moderately applicable in regional and global aerosol models at least for areas with the same average fleet composition as in our measurement site in Helsinki. However, more research is needed to obtain emission factors separated into the different types.

The annual $CO_2$ emission rate from traffic in Helsinki in 2017 was $5.38 \times 10^8\,\mathrm{kgCO_2 \cdot a^{-1}}$ (VTT Technical Research
Centre of Finland Ltd, 2017). Using the average NCA emission factor versus $CO_2$ emission, $2.21 \times 10^{14}\,\mathrm{kgCO_2^{-1}}$, a rough estimation on the annual NCA emission from traffic in Helsinki becomes $1.19 \times 10^{23}\,\mathrm{a^{-1}}$. The annual NCA formation rate via photochemical NPF in Helsinki can be approximated using estimates of nucleation rate, from 1 to 10 $\mathrm{cm^{-3}s^{-1}}$, NPF event day count per year, from 30 to 120 $\mathrm{a^{-1}}$, NPF duration, from 2 to 4 h, measured in a rural area in Hyytiälä, Finland (Dal Maso et al., 2005; Kulmala and Kerminen, 2008) and in an urban area in Helsinki (Hussein et al., 2008), the total area of Helsinki,
$214\,\mathrm{km^2}$, and a rough estimate for the boundary layer height, 500 m. Multiplying these gives the estimation of the annual NCA formation rate from $0.23 \times 10^{23}$ to $18.5 \times 10^{23}\,\mathrm{a^{-1}}$. Comparison of these annual rates suggests that in minimum of 6 % but even up to 84 % of NCA particles are estimated to originate from traffic in Helsinki on an annual basis. Although this range is wide, the contribution of traffic-originated NCA is significant.

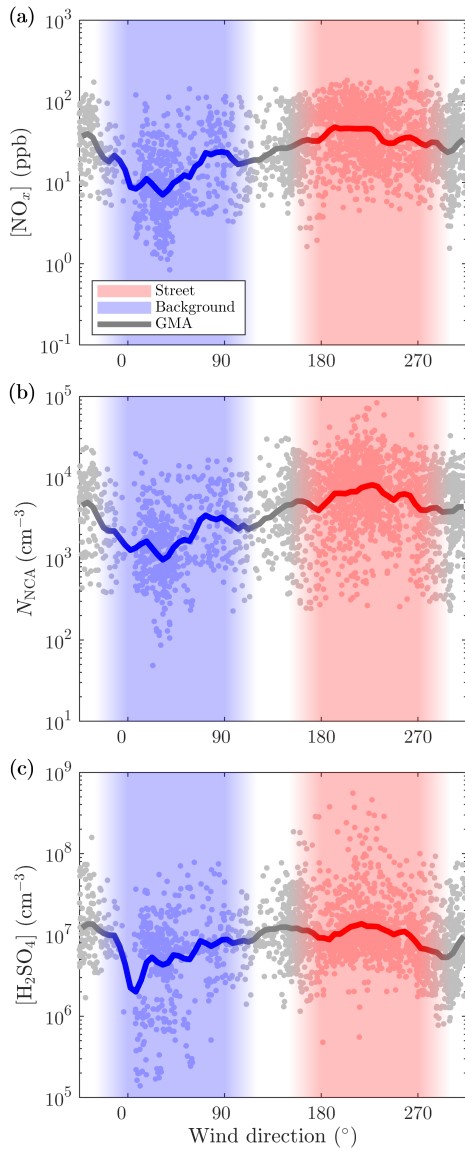

**Figure 8.** 10-min-averages of the (**a**) $NO_x$, (**b**) NCA, and (**c**) $H_2SO_4$ concentrations measured with different wind directions. Wind velocities smaller than 0.5 m/s are excluded. GMA denotes geometric moving average.

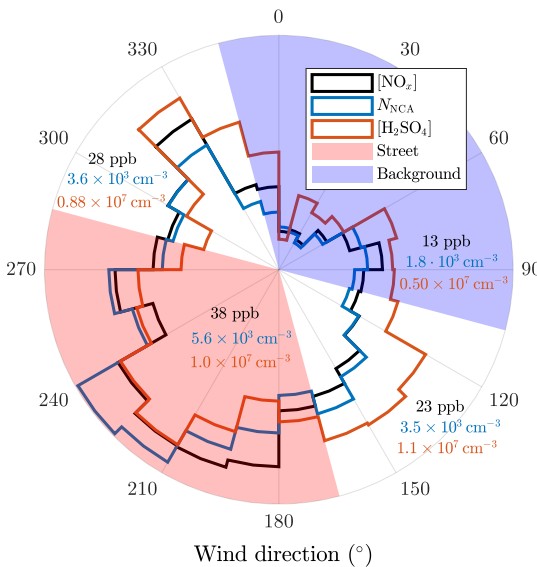

**Figure 9.** 10-min-averages of the $NO_x$, NCA, and $H_2SO_4$ concentrations further averaged to different wind direction sectors. Wind velocities smaller than 0.5 m/s are excluded. The geometric averages of the concentrations across the sectors are shown in the figure.

Another estimation for the traffic contribution on NCA (and also on $H_2SO_4$) in urban air can be performed using the linear fits from Fig. 6. Considering typical weekday noontime at our measurement location and assuming the annual mean of the daytime maximum SI in Helsinki, $500\,\mathrm{Wm}^{-2}$, the NCA concentration due to traffic is $\sim 9.7 \times 10^3\,\mathrm{cm}^{-3}$ (the value of the high $NO_x$ line at zero irradiance) and the increase of the NCA concentration due to photochemistry is $\sim 1.7 \times 10^3\,\mathrm{cm}^{-3}$ (calculated
with the slope of the low $NO_x$ line). These concentrations indicate that approximately 85 % of the total NCA concentration at the street canyon is originated from traffic at noontime. Considering midsummer and midwinter, the daytime maximum SI are $850\,\mathrm{Wm}^{-2}$ and $100\,\mathrm{Wm}^{-2}$, giving the contributions of $\sim$78 % and $\sim$97 %, respectively. Therefore, it is evident that the major fraction of NCA originated from traffic at our measurement location, even with the highest available SI values in midsummer.

For $H_2SO_4$, the concentration due to traffic at our measurement location at typical weekday noontime is $\sim 12 \times 10^6\,\mathrm{cm}^{-3}$
and the increase of the concentration due to photochemistry is $\sim 5.6 \times 10^6\,\mathrm{cm}^{-3}$, indicating approximately 68 % of the total $H_2SO_4$ concentration at the street canyon is originating from local traffic at noontime. For midsummer and midwinter, the contributions become $\sim$56 % and $\sim$92 %, respectively. These values signify that also the major fraction of $H_2SO_4$ originated from traffic even though it cannot be seen as clearly from the diurnal variation as is seen in the case of NCA.

Because regional NPF events are frequently suppressed by high condensation sinks (Kerminen et al., 2018), decreasing
condensation sink can lead to a NPF event, resulting in particle number concentration increase. However, our data display no clear anticorrelations of this kind (see Fig. S6). This again implies that regional NPF events cannot clearly be distinguished from the data measured in a vicinity of dense traffic.

Traffic-emitted NCA poses a potential health risk because the observed NCA concentrations are valid at the curbside of the street, which is the location where pedestrians spend time in traffic. Spreading of the NCA particles emitted on the streets can be approximated with particle lifetimes. The lifetimes can be estimated using coagulation sinks (CoagS) and the time constants of coagulation scavenging ($\tau_{CoagS}$), which is the inverse of CoagS, calculated as in Kulmala et al. (2001). Assuming no other losses of the particles, such as self-coagulation and condensational growth out from the NCA size range, and no mixing of the emitted aerosol with the background aerosol, $\tau_{CoagS}$ represents the lifetime of the particles. The estimated lifetimes were in the scale of several minutes, resulting in a spreading possibility of the NCA particles around urban areas. The diurnal variations of CoagS and $\tau_{CoagS}$ are presented in Fig. S2.

Our data clearly demonstrate that NCA-sized particle concentrations in a traffic-influenced environment is controlled by NCA directly emitted by traffic. The data also demonstrate that while generally NCA and photochemically produced nucleating vapor concentrations correlate, this correlation is likely, firstly, due to increased traffic volumes at daytime and, secondly, due to traffic-originated $H_2SO_4$ and other nucleating vapors. We also showed that $H_2SO_4$ formation is driven by both solar radiation and a traffic-related source.

## 4    Conclusions

Our results have several implications on our understanding of aerosol particle formation in traffic-influenced areas. Firstly, because current regional and global air quality models do not include particles in the sub-3 nm size range as primary emissions (Paasonen et al., 2016), the modelled NCA-sized particle concentrations are mainly driven by photochemical NPF processes, neglecting their origin from traffic as primary sources. Thus, our results show an urgent need to update these emissions. In light of our results, it seems evident that there will be areas in which direct emissions dominate the formation of new aerosol. A rough calculation gives that, on an annual basis, up to 84 % of NCA can originate from traffic in Helsinki; and according to the measured NCA concentrations, on typical weekday noontime, ~85 % of the total NCA concentration was contributed by traffic at our studied site. In wintertime, this contribution may reach ~97 % due to lower SI, which highlights the need for updating the annual particle formation cycles in the models. Secondly, our results also show that both traffic emission and regional NCA formation signals can be distinguished for the most of the times, and that traffic also influences the formation of $H_2SO_4$. Together with the findings of Yao et al. (2018), this presents a significant update on the particle formation mechanisms in urban areas. As illustrated in Fig. 4, the particle concentration is controlled by the interplay of the two processes, with varying importance depending on the proximity of the emission source. Our results call for reconsideration and re-analysis of observations of NPF events observed in traffic-influenced areas. In many cases, there is covariance between traffic volumes and SI, and care should be taken to separate these two variables in the analysis, e.g., by considering $CO_2$ or $NO_x$ as tracers for traffic volumes. Finally, potential health effects of traffic emissions in urban areas should also be considered more carefully because the composition of the emitted NCA particles is still unknown, especially as some clues for their non-volatility exist.

*Data availability.* The time series data are freely available at: (the link will be shown when the manuscript is accepted)

*Author contributions.* MD, TR, JVN, and HJT. designed the research. MO, HK, MA, JK, NK, and MI, performed the measurements. MO, HK, NK, and MI analyzed the data. MO prepared the manuscript with contribution from all co-authors.

*Competing interests.* The authors declare no conflict of interest.

5    *Acknowledgements.* We thank the tofTools team for providing tools for mass spectrometry analysis and Prof. Mikko Sipilä from the University of Helsinki for lending the chemical ionization inlet for the atmospheric pressure interface time-of-flight mass spectrometer. Dr. Harri Portin and Dr. Anu Kousa from Helsinki Region Environmental Services Authority (HSY) as well as the HSY's AQ measurement team are acknowledged for their valuable work related to the data quality control and measurements at the Mäkelänkatu supersite. Mr. Petri Blomqvist from the City of Helsinki is acknowledged for the traffic count data. The research has received funding from Tekes - the Finnish Funding
10   Agency for Innovation (Grant no. 2883/31/2015), HSY, and Pegasor Oy, who funded the research through the Cityzer project, the graduate school of Tampere University of Technology, and Academy of Finland for Profi 4 (Grant no. 318940) and infrastructure funding (Grant no. 273010).

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
