# Peer review of "Traffic-originated nanocluster emission exceeds $H_2SO_4$ -driven photochemical new particle formation in an urban area"

_Atmospheric Chemistry and Physics, 2019_

## Referee Comment (RC1) · Anonymous Referee #3 · 29 Oct 2019

General Comments: Olin et al. leverage previously collected nanocluster aerosol (NCA), trace gas (H2SO4, NOx, CO2), meteorological, and particle size distribution data from a month-long sampling campaign in Helsinki, Finland (Hietikko et al., 2018) to propose an updated model of NCA and H2SO4 formation in urban environments (Figure 4). This model is comprised of two pathways for generating NCA and H2SO4, respectively. Olin et al. argue that immediate implications from H2SO4-containing NCA on human health necessitates models including their H2SO4-NCA conceptual model. Generally, speaking none of the highlighted pathways are novel or new. Primary H2SO4 was identified by Arnold et al. (2012), and the direct emission of NCA from vehicles was identified by Rönkkö et al. (2017). To this end, the proposed model,

and ensuing discussions, feel more well suited for a review-type article opposed to a research article. This particularly true as much of the analysis focused on supporting that NCA sourced from traffic and not regional NPF events. This argument seemed redundant to the earlier paper (Hietikko et al., 2018) which the NCA data was sourced from.

The manuscript is generally well written, and the arguments are comprehensible. The manuscript's shortfall is the lack of NCA composition data. The lack of composition data makes the influence of primary $H_2SO_4$ to both number and mass concentration of vehicle-emitted NCA unsubstantiated and, from my perspective undermines what would be the major contribution of this manuscript (in respect to the model). The authors adequately demonstrate that NCA is decoupled from NPF events via a series of regression and correlation analyses. However, the reliance on data published in Hietikko et al. (2018) undermines the impact of these observations, as this was the primary focus of the earlier study. In contrast to the earlier study, Olin et al. do provide an off-the-hand annual estimate of traffic-derived NCA in Helsinki using $CO_2$ emission factors. I find that the general applicability of this approach might be questionable. In particular, the authors suggest the pathways (Figure 4) need consideration in regional chemical transport models but provide no clear means to facilitate this implementation. I can imagine that the complexity of this challenge would likely be confounded by varying relationships between NCA and $H_2SO_4$ and vehicle emissions (engine types, fuel types, emission standards, etc.) but am not enough of an expert to make specific recommendations. The challenges in implementing the conceptual model in regional chemical transport models is not elaborated on in this manuscript.

Lastly, I felt details pertaining to rationale and underlying assumptions were sometimes lacking in this manuscript. Specific points are made below. Although short manuscripts are ideal, I think this manuscript would greatly benefit from a more robust discussion on why certain decision were made. For instance, the authors opt to utilize $CO_2/NO_x$ as a proxy for traffic flow when the authors have direct measurements of traffic flow.

Why not just use traffic flow? This was particularly odd as the authors state that background fluctuations in CO2 make it an unreliable proxy for local traffic. I suspect these decisions might relate to the desire of making their observations relatable to chemical transport models and vehicle emission factors. There may be other reasons; however, they are not clearly stated in the manuscript.

Specific Comments: 1) As stated in the general comments, my most significant issue with the manuscript is that the composition of the NCA is not actually measured during the deployment in question. This is important as much of the manuscript relies on the argument that NCA composition is decoupled from secondary H2SO4, and thus, NCA should be compositionally influenced only by primary H2SO4 (Figure 4). To this point, the authors state on line 32, page 2: "the unknown chemical composition of traffic-originated NCA-sized particles...". Although, I do not view the author's conclusion as impossible, or even unlikely, as evidence exists that emissions from motor vehicles contain H2SO4 gas which may contribute NCA emissions (Arnold et al., 2012). Nonetheless, the authors fail to support their argument with results at hand. At present, the evidence suggests NCA formation can occur independent of NPF events in respect to the H2SO4 condensation sink (CS). I appreciate the challenge in measuring the composition of particles on a particle-by-particle basis or the small size range of NCA due to mass constraints; however, pathway 1A (figure 4) is not supported in their data.

2) From my understanding organic vapors, NH3, NOx, and H2SO4, are all precursor gases to new particle formation events (Kerminen et al, 2018). I am not an expert in NPF events and do not have a great sense of the relative occurrence and frequency that these different gases contribute to NPF events. At present, CS is only calculated in respect to H2SO4. I encourage the authors calculate CS in respect to trace gases known to contribute to NPF events. The argument that these are truly primarily emitted particles will be supported by a more robust calculation of CS.

3) Presently, it is unclear about the frequency of NCA events in respect to the CS.

Timeseries data for NCA and CS are included as supplementary figures. The authors' current presentation makes it difficult to distinguish events as there appears to be a lot of covariance between CS and NCA (and other diurnal properties). I encourage the authors to include these panels (as well as traffic flow and solar radiation) as a main figure. Furthermore, it would be nice if the authors provided a more quantitative feel for the frequency that NCA events occur during periods of high CS. There may be better ways to do it, but something along the lines of a running (windowed) Pearson's correlation (defining an R threshold as an event) between CS and NCA would prove helpful in distinguishing these periods.

4) The authors argue that NOx provides a good proxy for traffic flow. This claim is moderately supported by averaged trend data provided in Figure 3. My concerns are twofold. First, although the variance in the profiles appear to co-vary, why not simply correlate NOx and traffic profiles (separating weekends and weekdays). As a side note, I do not understand the decision to take the geometric mean opposed to the arithmetic mean. Second, the authors do not state the significance for using NOx as a proxy for traffic. It is strange to me as the authors have a direct measure of traffic 600 m up the road which they use for justifying NOx as a tracer for traffic.

5) Perhaps I am missing the rationale; however, I think Figure 8 should be SI vs CO2, where SI is not binned into separate groups. If the argument is that CO2 does not have a diurnal profile, then the slope will be $\sim 0$.

6) The abstract was lacking quantitative finding. I find that statements such as, "frequently correlated" (line 6, Page 1), to be extremely vague and really should not be in abstracts.

7) There is no statement on the availability of underlying data (https://www.atmospheric-chemistry-and-physics.net/about/data_policy.html).

8) The abstract makes mention of health effects. However, this was not a major finding in this work. Although fine to mention in the introduction and conclusion, it should not

be included in the abstract.

9) From my perspective, this is optional. The authors propose a conceptual model showing sink processes for NCA. I am generally interested in the loss rates for NCA in respect to the observed size distributions (i.e., coagulation rates). This would add a distinguishing feature from analysis presented in Hietikko et al. (2018). The authors would probably need to make assumptions about air parcels not mixing; however, this would provide an upper-bound estimate for NCA lifetime. I think this may have implications towards the authors earlier comments about potential health effects.

10) The authors did not outline how CS was calculated. Please include in the methods.

11) It is unclear what the authors mean by "weighting factor" as the authors regression analysis was not outlined in the methods. I am assuming the authors just average everything in a given bin.

12) The authors choice for bin widths (all regressions) may be justified but appear arbitrary without presenting a rationale. Personally, I think the data underlying regressions in Figures 6c,d Figure 7, and Figure 8 should be shown and not binned. If the data density is too high (graphical representation), the authors could possibly facet the different NOx and SI bins.

13) Figures 6c,d-8 should have confidence intervals for the intercepts so readers can evaluate the robustness in the intercepts (really relevant to 6c).

14) Line 18, Page 9: How many instances of NCA events occurred during cloudy weather? Was this the only time? This could be highlighted on the timeseries mentioned in main comment 3.

15) Line 2, Page 10: I am not familiar with momentary concentrations.

16) Line 4, Page 10: The use of relating NCA/H2SO4 to CO2 appears be creating an annual estimate of NCA. I did not think this was clearly articulated. Furthermore, on line 28, Page 5, the authors state: "because traffic does not cause a clear signal on

the measured CO2 concentration due to a high and varying CO2 background level, the NOx concentration was selected to represent the traffic-originated emissions overall."

Technical Comments: 1) Line 21, Page 2: "an evidence" should just be "evidence"

2) Line 10, Page 17: two doi's
* * *

---

## Referee Comment (RC2) · Anonymous Referee #2 · 1 Nov 2019

The paper deserves publication in the journal as I had noted in the review of the first version of the manuscript. I appreciate the work done by the authors to convincingly address all comments, including mine and the comments of the other reviewers. They have clarified points that were unclear in the first version. The revised version flows nicely and brings out both the new information and the questions they raise. I think it will be a useful contribution to the literature of particle formation mechanisms in urban areas.

---

## Author Comment (AC1) · 19 Nov 2019

The final response can be found in the supplementary file.

Please also note the supplement to this comment:
https://www.atmos-chem-phys-discuss.net/acp-2019-458/acp-2019-458-AC1-supplement.pdf

———————————————————

---

## Author Response (AR1)

**Final response to the referees' comments for Olin et al.: "Trafficoriginated nanocluster emission exceeds $H_2SO_4$ -driven photochemical new particle formation in an urban area"**

We thank the referees for their insightful comments and have corrected the manuscript according to them.

Referee reports are in *black italic* and authors' response in blue roman font. The marked-up manuscript and the Supplement highlighting the changes are included at the end of this file.

**Referee #2 comments:**

The paper deserves publication in the journal as I had noted in the review of the first version of the manuscript. I appreciate the work done by the authors to convincingly address all comments, including mine and the comments of the other reviewers. They have clarified points that were unclear in the first version. The revised version flows nicely and brings out both the new information and the questions they raise. I think it will be a useful contribution to the literature of particle formation mechanisms in urban areas.

We thank the referee for reading the revised manuscript.

**Referee #3 comments:**

General Comments: Olin et al. leverage previously collected nanocluster aerosol (NCA), trace gas (H2SO4, NOx, CO2), meteorological, and particle size distribution data from a month-long sampling campaign in Helsinki, Finland (Hietikko et al., 2018) to propose an updated model of NCA and H2SO4 formation in urban environments (Figure 4). This model is comprised of two pathways for generating NCA and H2SO4, respectively. Olin et al. argue that immediate implications from H2SO4-containing NCA on human health necessitates models including their H2SO4-NCA conceptual model. Generally, speaking none of the highlighted pathways are novel or new. Primary H2SO4 was identified by Arnold et al. (2012), and the direct emission of NCA from vehicles was identified by Rönkkö et al. (2017). To this end, the proposed model, and ensuing discussions, feel more well suited for a review-type article opposed to a research article. This particularly true as much of the analysis focused on supporting that NCA sourced from traffic and not regional NPF events. This argument seemed redundant to the earlier paper (Hietikko et al., 2018) which the NCA data was sourced from.

We agree and apologize that the current form of the manuscript does not highlight our new findings clearly enough. It is true that the direct NCA emissions from vehicles have already revealed by Rönkkö et al. (2017). We performed a new measurement campaign in Helsinki in 2017 and the NCA concentrations measured in this campaign were published by Hietikko et al. (2018). However, in the current manuscript we present, in addition to the NCA measurement data, also  $H_2SO_4$  and solar irradiance data, providing new and unpublished data and analysis that presents a novel interpretation of the data.

According to our understanding of the referee's criticism, the referee has understood our proposed model in a way that we claim that NCA contains and is formed solely from primary  $H_2SO_4$ . However, this is not the purpose of our claim, and we agree that this was not expressed clearly enough. Our main message is that NCA in an urban environment is, mainly, not formed by secondary  $H_2SO_4$ , which does not imply that NCA is formed from primary  $H_2SO_4$  solely. Instead, other routes, such as primary (possibly solid) emissions from vehicles, can explain the presence of NCA. This primary emission route was clearly missing from the referee's interpretation of our proposed model and is thus now clarified in the manuscript.

As far as  $H_2SO_4$  emitted directly by vehicles, the route already identified by Arnold et al. (2012), our novel finding is the secondary route (includes solar radiation) of  $H_2SO_4$  from vehicles. Additionally, instead of measuring  $H_2SO_4$  directly from an engine (Arnold et al., 2012), our measurements are to our knowledge the first field measurements connecting urban  $H_2SO_4$  concentrations to traffic sources quantitatively, and are therefore novel research. Our data measured at a curbside of a street provide reference data for primary  $H_2SO_4$  emission data and the ability to determine emission factors of vehicles in a real-world driving situation, for the first time. With the fact that vehicles emit  $H_2SO_4$  directly but our measurement at the curbside

displays no clear increase in  $H_2SO_4$  concentrations with increasing traffic volumes, it can be concluded that other routes for primary  $H_2SO_4$  emissions must exist. The most probable routes are related to gas-to-particle conversion, i.e. nucleation and condensation (routes 1A and 1B in Fig. 4). Thus, primary  $H_2SO_4$  mainly terminates from the gas phase either by nucleating to new particles (1A) or by condensing onto existing particles (1B). However, we cannot identify the ratio of these two routes and thus do not claim that NCA contains or is formed from  $H_2SO_4$  but do not rule that out either. Due to the fact that the primary  $H_2SO_4$  and nucleation mode particle number concentrations in vehicle exhaust are correlated (Arnold et al., 2012; Rönkkö et al., 2013), it is likely that the nucleation route (1A) does exist. Additionally, because the sizes of the nucleation mode particles are also correlated with the primary  $H_2SO_4$  concentrations (Rönkkö et al., 2013), it is likely that the condensation route (1B) exists as well. Because the ratio of the contribution of primary NCA emissions and the  $H_2SO_4$  nucleation route (1A) is also not identified, the relative proportion of  $H_2SO_4$  in the composition of the NCA particles is unknown. The text related to these is now clarified in the manuscript.

The manuscript is generally well written, and the arguments are comprehensible. The manuscript's shortfall is the lack of NCA composition data. The lack of composition data makes the influence of primary H2SO4 to both number and mass concentration of vehicle-emitted NCA unsubstantiated and, from my perspective undermines what would be the major contribution of this manuscript (in respect to the model). The authors adequately demonstrate that NCA is decoupled from NPF events via a series of regression and correlation analyses. However, the reliance on data published in Hietikko et al. (2018) undermines the impact of these observations, as this was the primary focus of the earlier study. In contrast to the earlier study, Olin et al. do provide an off-the-hand annual estimate of traffic-derived NCA in Helsinki using CO2 emission factors. I find that the general applicability of this approach might be questionable. In particular, the authors suggest the pathways (Figure 4) need consideration in regional chemical transport models but provide no clear means to facilitate this implementation. I can imagine that the complexity of this challenge would likely be confounded by varying relationships between NCA and H2SO4 and vehicle emissions (engine types, fuel types, emission standards, etc.) but am not enough of an expert to make specific recommendations. The challenges in implementing the conceptual model in regional chemical transport models is not elaborated on in this manuscript.

We agree that the NCA composition data would have been very beneficial, but due to a lack of suitable technology on measuring that directly because of very small particle sizes, the exact composition remains unknown. Examining the formation mechanisms of particles provides an alternative way for getting clues on their compositions, which is done in this manuscript. This is now mentioned in the manuscript. So far, our results imply that the secondary routes (including solar radiation) from vehicle-originated NCA formation mechanisms are not dominant, but additional research is needed to get more clues.

The annual estimation of the contribution of traffic and regional NPF on the presence of NCA in urban air was a rough estimation only in order to obtain the significance of traffic on NCA loadings. The text related to this is now modified to highlight the roughness of the estimation.

The challenges in implementing our model in regional chemical transport models (CTMs) include the effects of vehicle, engine, fuel, and after-treatment system types on  $H_2SO_4$  and NCA emission factors, which can vary markedly between the different types. Nevertheless, our emission factors represent the average fleet-level emission factors and are thus moderately applicable in regional CTMs at least for areas with the same average fleet composition as in our measurement site in Helsinki. More research is needed to obtain emission factors separated into the different types, but it is not an objective of this manuscript. Neither are the emission factors intended to act as quantitative emissions factors for regional CTMs, but are intended to distinguish the features of solar radiation and traffic levels. However, the results show that these processes need to be distinguished in modelling studies to obtain realistic results. These views are now added to the manuscript.

Lastly, I felt details pertaining to rationale and underlying assumptions were sometimes lacking in this manuscript. Specific points are made below. Although short manuscripts are ideal, I think this manuscript would greatly benefit from a more robust discussion on why certain decision were made. For instance, the authors opt to utilize CO2/NOx as a proxy for traffic flow when the authors have direct measurements of traffic flow. Why not just use traffic flow? This was particularly odd as the authors state that background fluctuations in CO2 make it an unreliable proxy for local traffic. I suspect these decisions might relate to the desire of making their observations relatable to chemical transport models and vehicle emission factors. There may be other reasons; however, they are not clearly stated in the manuscript.

The reason for using  $NO_x$  and  $CO_2$  as traffic tracers is twofold. Firstly, traffic density was measured in only 15 min time resolution, but the  $NO_x$  and  $CO_2$  data are available in 1 min resolution, which is the shortest resolution used in the analysis. Secondly, the flow field at the street canyon is turbulent with variable wind directions; thus, measured air parcels were diluted in a significantly different extent. Fortunately, the problem of different dilution ratios can be solved by using an exhaust-originated tracer because it has been diluted in the same extent as the emission of interest.

An ideal tracer is one that is universally emitted by all vehicles and is not altered in the atmosphere during the time scale of the exhaust plume dilution process.  $CO_2$  is emitted by all combustion engines, with the emission rate proportional to the fuel consumption; additionally, it remains unchanged in the atmosphere. The drawback of  $CO_2$  is its varying background concentration due to regional-level phenomena. The background concentration varied between 405 and 420 ppm during this measurement campaign; however, the effect of traffic on the  $CO_2$  concentration levels at the curbside of the street canyon was only 50 ppm in maximum. Thus, the maximum deviation in the background concentration (15 ppm) compared to the traffic contribution is significant. On the other hand, as the main source of  $NO_x$  in urban areas is traffic (Clapp and Jenkin, 2001), the background concentration is low and causes thus no significant uncertainty to the traffic contribution. The ratio of the two components of NOx (NO and NO2) is altered in atmospheric oxidation, but the sum (NOx) remains constant (Clapp and Jenkin, 2001). The drawback of  $NO_{\tau}$  as a tracer is its varying emission rates across the whole vehicle fleet (Yli-Tuomi et al., 2005). Concluding, we decided to use  $NO_r$  as the traffic tracer in our analysis, except in the emission factor analysis where we used  $CO_2$  because it is directly connected to the fuel consumption. Fuel consumption-related emission factors are practicable because they can provide estimations on total regional emissions and because they are the most understandable by the general audience. As observed from Fig. 4, a linear behavior of the concentrations is seen when averaged to  $CO_2$  concentration bins although the variance in the background concentration is an issue. As a comparison, Fig. FR1 presents examples of these graphs if the binning were done using traffic density or the  $NO_x$  concentration, and it can be seen that such a clear linear behavior is not obtained, especially in the case of traffic density. The clarification on the choice for traffic tracers is now added to the manuscript.

Figure FR1. Comparison of using (a,b) the  $CO_2$  concentration, (c,d) the  $NO_x$  concentration, or (e,f) traffic density as a binned quantity.

Specific Comments: 1) As stated in the general comments, my most significant issue with the manuscript is that the composition of the NCA is not actually measured during the deployment in question. This is important as much of the manuscript relies on the argument that NCA composition is decoupled from secondary H2SO4, and thus, NCA should be compositionally influenced only by primary H2SO4 (Figure 4). To this point, the authors state on line 32, page 2: "the unknown chemical composition of traffic-originated NCA-sized particles. ...". Although, I do not view the author's conclusion as impossible, or even unlikely, as evidence exists that emissions from motor vehicles contain H2SO4 gas which may contribute NCA emissions (Arnold et al., 2012). Nonetheless, the authors fail to support their argument with results at hand. At present, the evidence suggests NCA formation can occur independent of NPF events in respect to the H2SO4 condensation sink (CS). I appreciate the challenge in measuring the composition of particles on a particle-by-particle basis or the small size range of NCA due to mass constraints; however, pathway IA (figure 4) is not supported in their data.

**Please refer to the first reply.**

2) From my understanding organic vapors, NH3, NOx, and H2SO4, are all precursor gases to new particle formation events (Kerminen et al, 2018). I am not an expert in NPF events and do not have a great sense of the relative occurrence and frequency that these different gases contribute to NPF events. At present, CS is only calculated in respect to H2SO4. I encourage the authors calculate CS in respect to trace gases known to contribute to NPF events. The argument that these are truly primarily emitted particles will be supported by a more robust calculation of CS.

At least low volatile organic compounds,  $H_2SO_4$ ,  $NH_3$ , and amines are possible precursor gases to NPF (Kerminen et al., 2018). CS is in most cases calculated only for  $H_2SO_4$  (Lehtinen et al., 2007) and the NPF frequency quantified with respect to CS for  $H_2SO_4$  only, even though  $H_2SO_4$  would have not been measured. CS was calculated using the function (Kulmala et al., 2001):

$$CS = 2\pi \mathcal{D} \int_{0}^{\infty} \beta(D_{\rm p}) \cdot D_{\rm p} \cdot \frac{\mathrm{d}N}{\mathrm{d\log}D_{\rm p}} \mathrm{d\log}D_{\rm p}$$
(FR1)

where D is the diffusion coefficient of the condensing vapor and  $\beta(D_p)$  is the transition regime correction factor, which also depends on the condensing vapor. Choosing a different condensing vapor in calculating CS adds mainly a constant multiplier to the CS time series calculated for H2SO4, the multiplier depending on the molecular size of the vapor. Figure FR2 presents examples of the CS diurnal variations calculated for different vapors and it is now added to the Supplement.